# Parents of Adolescents with Anorexia Nervosa and Parents of Adult Women with Anorexia Nervosa

**DOI:** 10.3390/nu17071115

**Published:** 2025-03-24

**Authors:** Federico Amianto, Giulia Dell’Oca, Daniele Marcotulli, Chiara Davico, Andrea Martinuzzi

**Affiliations:** 1Neurosciences Department, University of Torino, Via Cherasco 15, 10126 Torino, Italy; giulia.delloca@edu.unito.it; 2Section of Child and Adolescent Neuropsychiatry, Department of Public Health and Pediatric Sciences, University of Turin, 10126 Torino, Italy; daniele.marcotulli@unito.it (D.M.); chiara.davico@unito.it (C.D.); andrea.martinuzzi@unito.it (A.M.)

**Keywords:** AN adolescents, age of onset, age of intake, mothers, resourcefulness, inadequacy, depression

## Abstract

**Background:** Anorexia nervosa (AN) remains a severe psychiatric disorder with multifactorial pathogenesis and an uncertain prognosis. It is essential to identify any factors that may contribute to its onset in order to improve the targeting of preventive and therapeutic interventions. The present study explores the characteristics of parents of daughters with AN, comparing those with different ages of onset and ages at intake, in order to identify potential contributing factors from the parental side. **Methods:** The study recruited 69 adolescents and 56 adults diagnosed with AN, along with 73 healthy controls (HC). The 80 mothers and 69 fathers of the participants were compared to controls in terms of the age of onset and age at intake of their daughters. Self-administered instruments were used to assess personality (TCI), eating behaviors (EDI-2), general psychopathology (BDI), and family functioning (FAD). **Results:** The analysis revealed that several personality and psychopathological traits distinguished the parents of AN participants from the HC group in both conditions. Both mothers of adolescent- and adult-onset AN participants with any age at intake displayed greater harm avoidance and lower self-directedness facets, as well as greater inadequacy and social insecurity than controls. Fathers were less disordered and more self-indulgent and compassionate, but lower in role definition. Specifically, mothers of daughters with adolescent-onset AN and younger age at intake were more prone to depression. In contrast, mothers of daughters with an adult age at intake showed lower resourcefulness and higher levels of perceived inadequacy. **Conclusions:** Higher resourcefulness and a lower sense of ineffectiveness may help mothers facilitate earlier therapeutic intervention for their daughters. On the other hand, maternal depressive symptoms may play a significant role in the earlier onset of and intake for AN in their daughters. Supporting assertive qualities in mothers through preventive interventions is recommended, while maternal depression should be appropriately treated to prevent an early psychopathological onset in daughters.

## 1. Introduction

Anorexia nervosa is a severe mental disorder that typically begins in adolescence or early adulthood but often persists or recurs later in life (DSM-5). It displays a complex and multifactorial pathogenesis that remains largely unknown [1]. However, it is widely believed that the family environment and its relational dynamics play a key role in the onset, development, and persistence of eating disorders (EDs) [2,3].

Recent research has shifted focus from exploring parental traits as risk factors for the onset of anorexia nervosa (AN) in daughters and sons. This is partly because it has been clearly established that no single specific parental profile can be definitively linked to the disorder [4]. Moreover, the correlation between parents’ personality traits and the psychopathology of daughters with AN is complex, making it difficult to draw definitive conclusions about the pathogenic role of parents [5]. Nevertheless, research on risk factors related to parental features is still of interest to facilitate family interventions.

In fact, research supports the importance of family involvement in therapeutic programs for AN [6]. Family involvement can include encouraging parents to restore their parenting roles with the support of the therapeutic team [7]. It may also involve parental counseling to enhance empathy and address dysfunctional family dynamics, which can improve therapeutic outcomes [8]; for instance, low levels of care from both parental figures have been linked to poorer therapeutic outcomes in patients with AN [9].

Parents’ personality traits interact with those of their daughters, forming the foundation of family dynamics which, in turn, influences the personality traits and eating psychopathology of daughters with AN [10]. This relationship is often nonlinear, involving complex interactions, as parental personality traits do not exert a singular and direct effect on their daughters’ psychopathology [5]. On one hand, the complexity of family dynamics is responsible for the inconclusive results of previous research. On the other hand, there is a call for caution when conducting and interpreting research on parents’ personality traits. Investigating these traits and their correlations with clinical and psychopathological features in daughters may inform counseling, family therapy, or even preventive interventions. For example, early studies using the Temperament and Character Inventory [10,11,12] showed that, compared to controls, mothers and fathers of patients with anorexia exhibit distinctive temperament and character traits. Fathers of patients with AN scored higher on harm avoidance (HA) and reward dependence (RD) and lower on persistence (P) and self-directedness (SD), while mothers scored higher on HA and lower on SD. These results were based on small samples and require replication, underscoring the need for more in-depth research [4,10]. Nevertheless, they show a relationship between parental personality profiles and the psychopathological features of daughters with AN [5].

Berking [13] recently underlined that psychopathology during adolescence may be conceived as a maladaptive strategy to cope with undesired affective states, and Ratcliffe [14] presented evidence that external “scaffolding” of emotions may be used by adolescents for emotional regulation, in particular when the regulatory structure is lost. Thus, parents represent a resource for emotional regulation that may be lost when psychopathology emerges in children. Family interventions based on counseling [8] or family therapy [15,16] are crucial to improve emotional regulation among adolescents affected with anorexia nervosa; this is even more relevant when parents display any kind of psychopathology [17].

Some studies examining the age of onset of AN have found that earlier therapeutic intervention during adolescence leads to a better prognosis than intervention in adulthood. Specifically, targeted psychotherapeutic and pharmacological treatments can reduce the risk of chronicity of this disorder’s severe psychopathological aspects [18,19]. However, an earlier age of onset may have different effects on the psycho-endocrine state [20] and the development of personality traits [21], and has been associated with worse long-term outcomes in terms of both psychiatric comorbidities and BMI [22]. Therefore, the age of onset and the timing of therapeutic intervention for AN appear to be important clinical indicators when choosing the most appropriate therapeutic approach [23].

Nevertheless, to the best of our knowledge, no study has explored the relationship between parental characteristics and the age of onset or intake for the treatment of AN. Thus, it remains unclear whether parents’ personality or psychopathological traits are related to the age of onset or intake for the treatment of AN in their daughters.

This study aimed to investigate the personality and psychopathological traits of parents of adolescent or adult daughters with AN. The first aim was to confirm previous findings on the specific personality and psychopathological traits that characterize parents of daughters with AN in these two distinct age groups and to compare these two groups of parents with healthy controls. The second aim was to explore whether these traits relate to the age of onset or intake for the treatment of AN among daughters. Given the effectiveness of family involvement in the therapeutic process of AN, linking the age of onset to parental features may have significant implications for preventive and therapeutic strategies [24,25].

## 2. Materials and Methods

### 2.1. Sample Description

The initial sample consisted of 118 female adolescents diagnosed with an ED who were referred to the outpatient services for EDs at the Child Neuropsychiatry Department of the Regina Margherita Children’s Hospital (OIRM), which is part of the City of Health and Science in Turin.

The inclusion criteria for parent–patient pairs were as follows: (1) a diagnosis of anorexia nervosa (AN) according to the DSM-5 (Diagnostic and Statistical Manual of Mental Disorders, Fifth Edition) criteria; (2) age under 18 years; (3) willingness of at least one parent to participate in the study; (4) the ability and willingness of both parent and adolescent to comprehend and complete self-administered tests; and (5) completion of the administered psychometric tests.

Based on these criteria, the following participants were excluded from the study: 13 individuals with binge eating disorder (BED), including 4 participants who also had obesity; 16 individuals with bulimia nervosa (BN); 6 individuals with obesity; 5 individuals with Otherwise Specified Feeding and ED (OSFED) who did not meet the criteria for atypical anorexia nervosa; 5 individuals with somatoform disorders; and 12 individuals with other diagnoses. Additionally, among the parents of the included participants, one mother and five fathers who were unavailable or who declined participation were excluded.

Thus, the final sample for this study consisted of parents (80 mothers and 69 fathers) of 80 adolescents with AN (Table 1). Two control groups of parents were recruited: 56 pairs of parents of adult participants with AN (56 mothers and 54 fathers) and 63 parents of healthy controls from the database of the Neuroscience Department of the University of Turin.

### 2.2. Procedure

The diagnoses of AN and atypical AN were established by an experienced neuropsychiatrist (FA) through a structured clinical interview (K-SADS) based on the DSM-5 criteria.

The severity of the disorder was assessed by the clinician using the Clinical Global Impression (CGI) scale at T0 (patient intake).

Parents of adolescent participants with AN completed the following battery of self-administered questionnaires: the Temperament and Character Inventory (TCI), the Eating Disorder Inventory-2 (EDI-2), the Beck Depression Inventory-II (BDI-II), and the Family Assessment Device (FAD).

### 2.3. Measures

#### 2.3.1. Clinical Global Impression [26]

The CGI Severity Scale is a scale which clinicians use to provide an estimate of the severity of a patient’s illness and psychopathology. The score is assigned by a psychiatrist-trained specialist, taking into consideration various aspects of the patient and their pathology: BMI, frequency and severity of binge eating and purging behaviors, overall functioning, and duration of illness.

#### 2.3.2. Temperament and Character Inventory [27]

The TCI is a self-report questionnaire based on the assumptions of the biosocial theory of personality developed by Cloninger [28]. It consists of 240 true/false items that evaluate personality features. Of these, 116 assess temperamental traits (novelty seeking [NS], HA, RD, and P); 119 evaluate three character traits (SD, cooperativeness [C], and self-transcendence [ST]), and 5 evaluate the presence of personality disorders (PDs).

#### 2.3.3. Eating Disorder Inventory-2 [29]

The EDI-2 test is a self-administered questionnaire that comprehensively assesses the salient features of psychopathology in ED. It consists of 91 items divided into 11 subscales: 3 subscales measure eating behavior (drive for thinness, bulimia, body dissatisfaction), while the other 8 subscales measure the psychopathological traits most frequently associated with ED (inadequacy, perfectionism, interpersonal distrust, interoceptive awareness, maturity fear, asceticism, impulsivity, and social insecurity).

#### 2.3.4. Beck Depression Inventory (BDI) [30]

The BDI is a self-administered questionnaire with 13 items that has been used extensively to assess the severity of depressive symptoms.

#### 2.3.5. Family Assessment Device [31]

The FAD is a self-administered questionnaire that assesses overall family functioning based on the perceptions of family members.

It is an assessment tool consisting of 60 items that investigate 7 different dimensions of family functioning: problem solving, communication, family roles, emotional resonance, emotional involvement, control behavior, and general operation.

### 2.4. Ethical Issues

This study was conducted in accordance with the Declaration of Helsinki as revised by the 75th World Medical Association General Assembly in October 2024, and in compliance with Good Clinical Practice (GCP) and current regulatory provisions. It was approved by the Intercompany Review Board of Torino (CEI) (protocol 00107/2019; protocol number 0099307).

### 2.5. Statistical Analysis

Although this was a retrospective study based on a convenience sample, we made certain our sample size would be adequate for hypothesis testing. Using the R package pwr, we computed the statistical power with an average group size of 55 participants to detect a medium effect size (Cohen’s f = 0.25) at a *p*-value of 0.05. The resulting power was 82%.

Statistical analyses were conducted using the advanced statistical software, Statistical Package for the Social Sciences (SPSS) version 28.0.1.0.

The personality and psychopathological traits of mothers and fathers of daughters with AN with adolescent onset/intake and those with adult onset/intake were compared with parents of healthy controls using the ANOVA test. These comparisons were conducted separately for parents based on their daughters’ age of onset and intake into therapeutic treatment. Due to the high number of comparisons, Bonferroni correction was applied, with the significance threshold set at *p* < 0.001.

## 3. Results

Table 1 displays the sociodemographic data of the adolescent participants.

**Table 1 nutrients-17-01115-t001:** Comparison of demographic and clinical data.

		Adolescent Patients (N = 69)	Adult Patients (N = 56)
**Age of onset of AN (years)**	mn ± sd	12.9 ± 2.9	17.9 ± 3.37
**BMI at intake**	mn ± sd	17.8 ± 2.8	16.9 ± 2.3
**CGI**	mn ± sd	4.4 ± 0.8	3.9 ± 1.2

BMI =  body mass index, CGI = clinical global impression.

### 3.1. Differences in Parents’ Traits Based on Age of Onset of AN in Daughters

#### 3.1.1. Mothers

Table 2 shows the ANOVA comparison between the mothers of daughters with AN onset in adolescence or adulthood and healthy controls.

The mothers of participants with adolescent and adult onset of AN showed higher levels of HA, particularly higher levels of fear of uncertainty (HA2), and lower levels of SD, particularly lower levels of maturity (SD1), resourcefulness (SD2), and purposefulness (SD3) (*p* < 0.001). They also displayed higher levels of social insecurity (IS) and inadequacy (IN) than healthy controls (*p* < 0.001).

The mothers of participants with AN onset in adolescence showed higher levels of depression (BDI) than healthy controls (*p* < 0.001).

#### 3.1.2. Fathers

Table 3 shows the ANOVA comparison between the fathers of daughters with AN onset in adolescence or adulthood and the fathers of healthy controls.

The fathers of participants with AN onset in adolescence or adulthood showed higher levels of self-acceptance (SD4) and compassion (C4) and lower levels of disorder (NS4) than the fathers of healthy controls (*p* < 0.001). They also displayed higher levels of role definition (FAD3) than the fathers of healthy controls (*p* < 0.001).

### 3.2. Comparison of Personality Traits of Parents of Daughters with AN Based on Age at Treatment Intake

Table 4 shows the ANOVA comparison of the personality traits of mothers of participants with AN, grouped according to their daughters’ age at treatment intake.

The mothers of adult participants at treatment intake showed higher levels of inadequacy (IN, *p* < 0.001) than the healthy controls, and higher levels of SD3 (*p* < 0.001) than both the healthy controls and those of adolescent participants at treatment intake.

In contrast, the mothers of adolescent participants at treatment intake showed higher BDI scores than the healthy controls (*p* < 0.001).

Both the mothers of adolescent participants and those of adult participants at treatment intake showed higher fear of uncertainty (HA2) and HA scores and lower SD1, SD2, and SD scores than the healthy controls (*p* < 0.001). They also displayed higher levels of IS than the healthy controls (*p* < 0.001).

Table 5 shows the ANOVA comparison of the personality traits of fathers of participants with AN, grouped according to their daughters’ age at treatment intake.

In particular, the fathers of adult participants at treatment intake showed higher levels of affective involvement (FAD5) than the fathers of healthy controls (*p* < 0.001). Both the fathers of adolescent participants and those of adult participants at treatment intake showed higher levels of SD4 and C4 and lower levels of NS4 than the fathers of healthy controls. They also displayed higher levels of FAD3 than the fathers of healthy controls (*p* < 0.001).

## 4. Discussion

The present study aimed to compare the personality and psychopathological traits of mothers and fathers of adolescents and young women affected by AN, grouped based on the age of onset of their daughters’ disorder and the age at intake to an outpatient service. Due to the cross-sectional nature of this study, no causal relationships can be established. Nevertheless, the data obtained can provide insights to support parental counseling for the prevention and treatment of this disorder [8].

In general, several differences in personality traits were found between the parents of participants with AN and the parents of healthy controls, confirming previous findings on this topic, which was the first aim of the study [10,11,12,32,33]. Mothers of participants with AN were characterized by high levels of harm avoidance and low levels of self-directedness in terms of personality traits, as well as a greater sense of inadequacy and social insecurity related to eating psychopathology. Moreover, most of these differences were common across subgroups of parents whose daughters were adolescents or adults at the onset of AN or at intake for treatment of AN.

Regarding the second aim of the study, few differences specifically distinguished the parents of participants with AN and the parents of healthy controls based on the age of onset of the disorder. Three differences emerged among mothers, while only one was observed among fathers. This may highlight the relatively greater influence of mothers on the onset of the disorder [34], but also potentially suggests fathers’ lower engagement in managing their daughters’ health problems, especially in consideration of the lower participation of fathers in this research.

The mothers of adolescent participants at treatment intake exhibited higher levels of resourcefulness than the mothers of adult participants at treatment intake. This may indicate greater problem solving ability, meaning they are better equipped to acquire the tools necessary to address challenges in their daughters’ upbringing [35]. In the general population, high levels of resourcefulness—a subdimension of self-directedness—are associated with a lower incidence of depressive [36] and other mental disorders [4]. The present study suggests that, while resourcefulness may not be protective against the onset of AN in daughters, it characterizes mothers who are more proactive and prepared to handle problematic situations, facilitating earlier intake to treatment services. Previous research suggests that early treatment intake is a positive prognostic factor for AN [19,37]. Thus, promoting positive attitudes towards treatment services among mothers may lead to better outcomes for their daughters.

Compared to the healthy controls, only the mothers of adult participants at treatment intake exhibited higher levels of inadequacy (EDI). The initiation of treatment for AN in adulthood may sometimes result from diagnostic delays [19,38], which could lead to feelings of guilt among mothers for failing to recognize the disorder in their daughters earlier, contributing to their sense of inadequacy [39]. Alternatively, perceived inadequacy may be associated with lower levels of resourcefulness in this group of mothers, potentially delaying their daughters’ treatment due to their lack of confidence in managing the issue [7]. Encouraging mothers to take the initiative in addressing their daughters’ illness through psychoeducation, promoting greater awareness of available therapeutic resources, and providing support in decision-making may help mitigate feelings of inadequacy and improve secondary prevention strategies [6,40,41].

The mothers of participants who were adolescents at the onset of AN or at treatment intake exhibited higher levels of depressive symptoms than the mothers of the control group. Various studies have highlighted that parental psychopathology, particularly in mothers, is strongly linked to the risk of EDs in daughters [42,43]. Elevated maternal depressive symptoms may, therefore, be considered a risk factor for earlier onset and, consequently, earlier treatment intake of AN in daughters. Increased maternal depression can negatively affect the entire family environment, increasing the stress and emotional burden felt by daughters with AN [44]. As evidenced by the literature on anxiety and depressive disorders, parents’ depression may impair the developmental trajectory of children before adulthood and represents a strong risk factor that may be associated with unfavorable parental behaviors [45,46].

Concerning fathers, no significant differences were found between subgroups based on the age of onset of AN or the age at treatment intake. While the existing literature indicates the role of fathers in the onset of AN [10,12,32,33], their influence may be less relevant to the age of onset of this disorder or the age at treatment intake compared to mothers.

In this study, fathers exhibited fewer personality differences compared with mothers. Fathers tend to be more disorganized, which may negatively impact daughters with obsessive–compulsive personality traits [47]. This temperamental trait may contribute to a weaker role definition within the family, potentially leading to dysfunctional family dynamics [10]. However, in this study, the fathers of participants with AN also displayed higher levels of self-acceptance and compassion compared to the fathers of healthy controls, which could serve as protective factors against the development of mental disorders [48].

The fathers of adult participants at treatment intake reported lower levels of emotional involvement compared to those of healthy controls, suggesting reduced participation in family activities and interests [49]. Previous studies have indicated that paternal parenting attitudes may predict daughters’ eating psychopathology [33,50]. A disengaged and passive father who demonstrates little involvement in his daughter’s life may contribute to an earlier onset of eating disorder symptoms [49,50,51].

Given the cross-sectional nature of this study and the relatively small cohort of families with daughters affected by AN, further research using prospective study designs is needed to explore the causal links between parental characteristics and AN onset and outcomes in daughters.

Additionally, due to the complexity and heterogeneity of families affected by AN in terms of clinical severity and treatment pathways, larger sample sizes are necessary to yield more generalizable results and make more specific inferences regarding different parental subgroups.

Despite the detailed assessment conducted by the neuropsychiatrist and the reliability of the diagnostic criteria used, a potential limitation of this study is the difficulty in retrospectively identifying the exact age of onset of the disorder with sufficient accuracy.

Finally, due to the small number of participants with binge/purging AN in the sample, this study did not separately analyze the two diagnostic subtypes, which may be associated with distinct parental characteristics.

## 5. Conclusions

As established in previous research, the parents of participants with AN exhibited distinct personality traits compared with the parents of healthy controls, with certain characteristics differentiating mothers of adolescent participants versus those of adult participants at the onset of AN or at treatment intake.

The mothers of adolescent participants at treatment intake for AN demonstrated higher levels of resourcefulness and lower levels of inadequacy, which potentially facilitated their earlier recognition of their daughters’ problems. According to the literature, promoting resourcefulness and feelings of adequacy through preventive psychoeducational interventions may improve the prognosis and treatment efficacy for daughters with AN. Conversely, raising awareness among mothers, general practitioners, and mental health professionals about early treatment for maternal depressive symptoms may help prevent the onset of AN in adolescents.

Although fathers were less differentiated in terms of personality traits compared with mothers when grouped based on the age of onset of AN and the age at treatment intake, they exhibited lower levels of emotional involvement. Targeted interventions should aim to enhance fathers’ affective involvement to increase their awareness of their daughters’ distress and needs.

Future studies should focus on the prospective assessment of parental personality’s influence on psychopathology, eating disorder severity, and treatment outcomes in daughters with AN. Additionally, research should explore interactions between parental personality and other aspects of family functioning, such as family history, socioeconomic factors, couple dynamics, and parent–daughter relationships. Finally, future investigations should differentiate participants based on AN diagnostic subtypes, specifically distinguishing between the restrictive subtype, the binge-eating/purging subtype, and atypical forms.

## Figures and Tables

**Table 2 nutrients-17-01115-t002:** ANOVA comparison of mothers based on age of onset of daughters’ AN.

	Adolescent Onset AN (a, *n* = 105) Mean ± sd	Adult Onset AN (b, *n* = 31) Mean ± sd	Healthy Controls (c, *n* = 63) Mean ± sd	F	*p*	η^2^	Post Hoc Analysis
Fear of uncertainty (HA2)	5.2 ± 1.4	5.1 ± 1.7	3.9 ± 1.9	13.244	0.001	0.122	b, a > c
Harm avoidance (HA)	18.4 ± 5.8	18.8 ± 7.1	14.5 ± 5.9	10.082	0.001	0.095	b, a > c
Maturity (SD1)	5.6 ± 1.7	5.1 ± 2.0	6.9 ± 1.5	18.681	0.001	0.164	b, a < c
Purposefulness (SD2)	5.1 ± 1.9	4.7 ± 2.0	6.2 ± 1.4	11.810	0.001	0.110	b, a < c
Resourcefulness (SD3)	3.7 ± 1.5	2.9 ± 1.9	4.3 ± 1.0	11.017	0.001	0.103	b, a < c
Self-directedness (SD)	29.9 ± 7.6	27.9 ± 9.6	33.7 ± 7.6	8.647	0.001	0.083	b, a < c
Inadequacy (IN)	3.5 ± 4.4	4.8 ± 4.5	1.7 ± 3.0	7.133	0.001	0.076	b, a > c
Social insecurity (IS)	3.8 ± 3.9	4.5 ± 4.2	2.0 ± 2.5	7.611	0.001	0.081	b, a > c
Beck Depression Inventory (BDI)	9.0 ± 7.2	7.2 ± 6.1	3.9 ± 6.7	10.775	0.001	0.104	a > c

a = Adolescent-Onset AN; b = Adult-Onset AN; c = Healthy Controls.

**Table 3 nutrients-17-01115-t003:** ANOVA comparison of fathers based on age of onset of daughters’ AN.

	Adolescent Onset AN (a, *n* = 91) Mean ± sd	Adult Onset AN (b, *n* = 32) Mean ± sd	Healthy Controls (c, *n* = 57) Mean ± sd	F	*p*	η^2^	Post Hoc Analysis
Disorder (NS4)	3.6 ± 1.9	3.4 ± 1.9	5.1 ± 2.2	11.880	0.001	0.123	a, b < c
Self-acceptance (SD4)	7.5 ± 2.7	7.9 ± 5.8	5.4 ± 3.2	12.572	0.001	0.121	a, b > c
Compassion (C4)	7.5 ± 2.0	8.4 ± 3.2	6.2 ± 2.8	8.816	0.001	0.087	a, b > c
Role definition (FAD3)	2.1 ± 0.4	2.2 ± 0.4	1.8 ± 0.6	7.779	0.001	0.005	a, b > c

a = Adolescent-Onset AN; b = Adult-Onset AN; c = Healthy Controls.

**Table 4 nutrients-17-01115-t004:** ANOVA comparison of mothers based on age of intake of daughters’ AN.

	Adolescent Intake (a, *n* = 80) Mean ± sd	Adult Intake (b, *n* = 56) Mean ± sd	Healthy Controls (c, *n* = 63) Mean ± sd	F	*p*	η^2^	Post Hoc Analysis
Fear of uncertainty (HA2)	5.1 ± 1.4	5.2 ± 1.6	3.9 ± 1.9	13.236	0.001	0.122	b, a > c
Harm avoidance (HA)	18.6 ± 6.0	18.5 ± 6.4	14.5 ± 5.9	10.034	0.001	0.095	b, a > c
Maturity (SD1)	5.7 ± 1.6	5.1 ± 2.0	6.9 ± 1.5	19.894	0.001	0.172	b, a < c
Purposefulness (SD2)	5.2 ± 1.7	4.8 ± 2.1	6.2 ± 1.4	12.267	0.001	0.114	b, a > c
Resourcefulness (SD3)	3.7 ± 1.4	3.0 ± 1.8	4.3 ± 1.0	12.750	0.001	0.118	b < a, c
Self-directedness (SD)	30.4 ± 6.9	28.3 ± 9.4	33.7 ± 5.9	9.031	0.001	0.086	b, a < c
Inadequacy (IN)	3.4 ± 4.0	4.5 ± 4.9	1.7 ± 3.0	7.318	0.001	0.078	b > c
Social insecurity (IS)	3.7 ± 3.4	4.4 ± 4.6	2.0 ± 2.5	7.954	0.001	0.084	b, a > c
Beck Depression Inventory (BDI)	9.9 ± 7.7	6.9 ± 5.7	3.9 ± 6.7	13.059	0.001	0.123	a > c

a = Adolescent-Onset AN; b = Adult-Onset AN; c = Healthy Controls.

**Table 5 nutrients-17-01115-t005:** ANOVA comparison of fathers based on intake of onset of daughters’ AN.

	Adolescent Intake (a, *n* = 69) Mean ± sd	Adult Intake (b, *n* = 54) Mean ± sd	Healthy Controls (c, *n* = 57) Mean ± sd	F	*p*	η^2^	Post Hoc Analysis
Disorder (NS4)	3.6 ± 1.8	3.4 ± 2.0	5.1 ± 2.2	12.080	0.001	0.124	b, a < c
Self-acceptance (SD4)	7.2 ± 2.7	8.0 ± 2.6	5.4 ± 3.2	13.545	0.001	0.129	b, a > c
Compassion (C4)	7.4 ± 2.1	8.2 ± 2.6	6.2 ± 2.8	9.122	0.001	0.090	b, a > c
Role definition (FAD3)	2.1 ± 0.4	2.2 ± 0.3	1.8 ± 0.6	8.210	0.001	0.083	b, a > c
Affective involvement (FAD5)	1.9 ± 0.5	2.1 ± 0.3	1.7 ± 0.6	6.957	0.001	0.071	b > c

a = Adolescent-Onset AN; b = Adult-Onset AN; c = Healthy Controls.

## Data Availability

Raw data of the study are available upon reasonable request to the corresponding author.

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
