# Peer review of "Parents of Adolescents with Anorexia Nervosa and Parents of Adult Women with Anorexia Nervosa"

_nutrients, 2025, doi:10.3390/nu17071115_

Round 1

Reviewer 1 Report

Comments and Suggestions for Authors

I have read the manuscript entitled: “Parents of AN adolescents vs parents of AN adult women: implications of common and different features” and I find it very well. The aim of this study was to compare the personality traits and psychopathology of parents of adolescent and adult young women affected with anorexia nervosa (AN), divided into subgroups based on the age of disorder onset and age of admission to the outpatient clinic.
The Authors carefully examined potential predictors related to parenting behaviors that could be associated with the onset of the disorder or the initiation of its treatment course.

Weaknesses of the study:

-relatively small cohort group given the complexity and heterogeneity of families with daughters affected with AN

Main findings of the study:

-the need to promote appropriate attitudes, such as resourcefulness, in mothers of AN daughters, through preventive psychoeducational interventions, which may improve the prognosis and effectiveness of treatment of daughters with AN

-raising awareness not only of mothers but also of family doctors and mental health professionals about the early treatment of depressive symptoms in mothers, which may help prevent the occurrence of AN in their daughters during adolescence

-the need for interventions aimed at increasing the emotional involvement of fathers, familiarizing them and increasing their awareness of the suffering and needs of their daughters.

The work is certainly valuable, contains elements of novelty and should have a strong impact on the community. I found only a few shortcomings related to the use of abbreviations, e.g. line 106 - the lack of explanation of BN-BP, line 140 – the lack of explanation of PD, ..

Some abbreviations are introduced several times, while, when first introduced, in the rest of the text, the abbreviation should be used consistently, e.g. eating disorders (ED), harm avoidance (HA), etc..

In the caption for table 5 - there is a mistake – there is: “ANOVA comparison of mothers …”  - it should be “fathers

 I noticed incorrect formatting in the tables, and the meaning of the symbols "a", "b" and "c" was not explained.

Author Response

I have read the manuscript entitled: “Parents of AN adolescents vs parents of AN adult women: implications of common and different features” and I find it very well. The aim of this study was to compare the personality traits and psychopathology of parents of adolescent and adult young women affected with anorexia nervosa (AN), divided into subgroups based on the age of disorder onset and age of admission to the outpatient clinic.
The Authors carefully examined potential predictors related to parenting behaviors that could be associated with the onset of the disorder or the initiation of its treatment course.

Weaknesses of the study:

-relatively small cohort group given the complexity and heterogeneity of families with daughters affected with AN

As suggested by the referee #2 we performed statistical exploration to grant the robustness of the findings

The work is certainly valuable, contains elements of novelty and should have a strong impact on the community. I found only a few shortcomings related to the use of abbreviations, e.g. line 106 - the lack of explanation of BN-BP, line 140 – the lack of explanation of PD, ..

Some abbreviations are introduced several times, while, when first introduced, in the rest of the text, the abbreviation should be used consistently, e.g. eating disorders (ED), harm avoidance (HA), etc..

-The paper was revised and all abbreviations explained at their first use, moreover they have been used consistently along the manuscript acconding to the referee’s suggestions.

In the caption for table 5 - there is a mistake – there is: “ANOVA comparison of mothers …”  - it should be “fathers

-The caption has been corrected

 I noticed incorrect formatting in the tables, and the meaning of the symbols "a", "b" and "c" was not explained.

-The tables have been correctly formatted and the symbols explained

Reviewer 2 Report

Comments and Suggestions for Authors

The authors present a manuscript of interest that is within the scope of this Journal. However, the authors should improve important aspects:

-The title of the manuscript is not clear. The authors should be more direct.

-The summary of the manuscript should have more content.

-The introduction to the state of the art should contain updated references. The authors should justify the novelty of the study.

-In the methodology, the authors should justify the sample size with statistical methods.

-The authors should perform the statistical calculation of the statistical potential based on the sample size.

-The consensus references should be included in the methodology, in an updated form.

-The authors should improve the legend figures. The authors should include all the information.

-The discussion should compare existing studies and with other diseases. The authors should conduct an organized discussion.

-In the discussion, the authors should include and justify the translation.

-The conclusion should be translational and solidly based on the results.

-Authors should improve their use of English grammar with professionals.

Comments on the Quality of English Language

The English could be improved to more clearly express the research.

Author Response

-The title of the manuscript is not clear. The authors should be more direct.

The title has been changed according to the Editor’s indications

-The summary of the manuscript should have more content.

The summary has been enriched with more content.

-The introduction to the state of the art should contain updated references. The authors should justify the novelty of the study.

New references have been included to update the introduction.

-In the methodology, the authors should justify the sample size with statistical methods.
-The authors should perform the statistical calculation of the statistical potential based on the sample size.

Statistical calculation of the power of the study has been included

-The consensus references should be included in the methodology, in an updated form.

The authors included the consent (?) references in their updated form

-The authors should improve the legend figures. The authors should include all the information.

The authors revised and improved the table legends (no figures are included in the manuscript)

-The discussion should compare existing studies and with other diseases. The authors should conduct an organized discussion.

The authors better organized the discussion.

-In the discussion, the authors should include and justify the translation.

The meaning of this request is obscure, nevertheless any problem with translation has been resolved by professional English editing

-The conclusion should be translational and solidly based on the results.

The conclusion has been revised according to the results of the paper.

-Authors should improve their use of English grammar with professionals. The English could be improved to more clearly express the research.

The language has been revised by a professional editor.

Reviewer 3 Report

Comments and Suggestions for Authors

Parents of AN adolescent's vs parents of AN adult women: implications of common and different features

Revise the title

L14-15: revise

The present study explores the characteristics of parents of daughters with AN, comparing those with different ages of onset and ages at intake, in order to identify potential contributing factors from the parental side. The analysis revealed that several personality and psychopathological traits distinguished the parents of AN participants from the HC group in both conditions. Mothers of daughters with adolescent-onset AN and younger age at intake were more prone to depression. While mothers of daughters with an adult age at intake showed higher levels of perceived inadequacy. The authors concluded that higher resourcefulness and a lower sense of ineffectiveness may help mothers facilitate earlier therapeutic intervention for their daughters. Maternal depressive symptoms may play a significant role in the earlier onset and intake of AN in their daughters.

Be consistent with abbreviations throughout the paper.

L42-47: revise, since it has been clearly established that no specific profile of mothers or fathers can be definitively linked to the disorder, what is the purpose of doing this research.

Revise the introduction: some conflicting ideas were provided.

L84-86: Revise, the goal is to confirm previous findings in these two distinct age groups and explore whether these traits relate to the age of onset or therapeutic intake of AN in their daughters, comparing two groups of parents with healthy controls. Confirm based on what, the sentence is not clear.

Results are good, however tables needed to be checked for formatting. P value should be revised P<.001 or <.01, all zeros are not acceptable

All tables should have explanation of the abbreviations used. 

All questionnaires should be provided as a supplement. 

Discussion: Some information was provided earlier, avoid repetition of sentences.

"The present study aimed to compare the personality and psychopathology traits of 
mothers and fathers of adolescents and young women affected by anorexia nervosa 
(AN), subgrouped based on the age of onset of their disorder and the age at intake into an
outpatient service. The objective was to identify potential predictors related to parental 
behaviors that could be associated with the onset of the disorder or the initiation of its 
therapeutic course. However, due to the cross-sectional nature of this study, no causal 
relationships can be established. Nevertheless, these data may provide insights to sup-
port parental counseling in both the prevention and treatment of the disorder.

Comments on the Quality of English Language

The manuscript needs revision for English and Santense structure. 

Author Response

Be consistent with abbreviations throughout the paper. The abbreviations have been revised throughout the paper L42-47: revise, since it has been clearly established that no specific profile of mothers or fathers can be definitively linked to the disorder, what is the purpose of doing this research. The sentence has been revised and clarified Revise the introduction: some conflicting ideas were provided. The introduction has been revised to reduce confusion L84-86: Revise, the goal is to confirm previous findings in these two distinct age groups and explore whether these traits relate to the age of onset or therapeutic intake of AN in their daughters, comparing two groups of parents with healthy controls. Confirm based on what, the sentence is not clear. The sentence has been revised by the authors and by the English editing service Results are good, however tables needed to be checked for formatting. P value should be revised P<.001 or <.01, all zeros are not acceptable Tables have been formatted according to referee’s requests All tables should have explanation of the abbreviations used. The abbreviations have been explained in all tables All questionnaires should be provided as a supplement. Questionnaires have been provided as supplementary material Discussion: Some information was provided earlier, avoid repetition of sentences. We summarized the beginning of the discussion to avoid repetition of sentences "The present study aimed to compare the personality and psychopathology traits of mothers and fathers of adolescents and young women affected by anorexia nervosa (AN), subgrouped based on the age of onset of their disorder and the age at intake into an outpatient service. The objective was to identify potential predictors related to parental behaviors that could be associated with the onset of the disorder or the initiation of its therapeutic course. However, due to the cross-sectional nature of this study, no causal relationships can be established. Nevertheless, these data may provide insights to sup- port parental counseling in both the prevention and treatment of the disorder.

Round 2

Reviewer 2 Report

Comments and Suggestions for Authors

Accept in present form.

Reviewer 3 Report

Comments and Suggestions for Authors

Parents of adolescent young people with anorexia nervosa and parents of adult women with anorexia nervosa. Revised version

No further comments